# Pulmonary Hypertension-Related Interstitial Lung Disease: An Expert Opinion with a Real-World Approach

**DOI:** 10.3390/biomedicines13040808

**Published:** 2025-03-27

**Authors:** Rachel N. Criner, Mario Naranjo, Gilbert D’Alonzo, Sheila Weaver

**Affiliations:** Department of Thoracic Medicine and Surgery, Lewis Katz School of Medicine at Temple University Hospital, Philadelphia, PA 19140, USA; rachel.criner@tuhs.temple.edu (R.N.C.); mario.naranjo-tovar@tuhs.temple.edu (M.N.);

**Keywords:** interstitial lung disease, idiopathic pulmonary fibrosis, pulmonary hypertension, WHO group 3 pulmonary hypertension, nintedanib, pirfenidone, treprostinil

## Abstract

Great progress has been made in the treatment of pulmonary arterial hypertension (WHO group 1 PAH) over the past two decades, which has significantly improved the morbidity and mortality in this patient population. Likewise, the more recent availability of antifibrotic medications for interstitial lung disease (ILD) have also been effective in slowing down the progression of disease. There is no known cure for either of these disease states. When this combination coexists, treatment can be challenging. Interstitial lung disease is a heterogenous group of chronic inflammatory and/or fibrotic parenchymal lung disorders. A subset of patients with ILD, not related to connective tissue disease, can initially present with inflammatory-predominant disease which progresses to irreversible fibrosis. This population of patients is also at risk for developing pulmonary hypertension (PH) or World Health Organization (WHO) group 3 PH. This coexistence of ILD and PH is associated with early morbidity and mortality. The early identification, diagnosis, and treatment of this combination of ILD and PH is vital. Medications available for both ILD and PH require an individualized approach with the intention of slowing down disease progression. Referral to expert centers for clinical trials and transplant evaluation is recommended. The combination of PH-ILD can be challenging to diagnose and treat effectively. Patients require a thorough clinical evaluation to enable the most accurate diagnosis. A vital part of that evaluation is the early recognition of PH. Medications can help improve disease progression along with clinical trials that will further improve our gaps in knowledge.

## 1. Introduction

Interstitial lung disease (ILD) is a heterogenous group of chronic inflammatory and/or fibrotic parenchymal lung disorders. Although considered a rare disease of aging, at less than 1% of the United States population in 2019, prevalence increased by almost 20% from 2010 to 2019 [1]. About 40% of ILD patients with initially inflammatory-predominant disease will develop irreversible, progressive fibrosis [2]. The development of pulmonary fibrosis increases the risk for respiratory failure as well as death with a mean survival of 4 years without treatment [3,4]. Additionally troubling is the development of pulmonary hypertension (PH) in ILD, reported to occur in 14–86% of ILD patients, with prevalence increasing as fibrosis progresses [2,5]. When PH is associated with ILD, patients have worse functional capacity, more severe hypoxemia, impaired health-related quality of life, increased acute exacerbation risk, increased likelihood for hospitalization, and a threefold increase in mortality [6]. Therefore, early identification of ILD and concomitant PH are paramount for treatment and close monitoring to be promptly initiated.

## 2. Background

Interstitial lung disease develops from numerous etiologies, including autoimmune diseases, occupational or environmental exposures, genetic factors, medications, infections, and idiopathic causes [7]. Etiologies are often associated with specific imaging patterns on high-resolution computed tomography (HRCT) and histologic patterns on biopsy [7]. Once fibrosis predominates, ILD patients are classified as having pulmonary fibrosis and are further divided into idiopathic pulmonary fibrosis (IPF) or progressive pulmonary fibrosis (PPF). IPF requires the usual interstitial pneumonia pattern on HRCT or biopsy and has no identifiable cause [8]. Conversely, PPF requires the presence of fibrosis on imaging from either known or unknown causes, other than IPF, with two of the three following criteria over the past 12 months (1) progressive respiratory symptoms, (2) an absolute forced vital capacity decline ≥ 5% or absolute DLCO decline ≥ 10%, and/or (3) image progression [9].

ILD can lead to the development of World Health Organization (WHO) group 3 PH, defined as a mean pulmonary artery pressure >20 mmHg at rest, a pulmonary arterial wedge pressure ≤ 15 mmHg, and a pulmonary vascular resistance (PVR) >2 Wood units (WU) [10]. Severe PH requires a PVR > 5 WU; it is associated with worse prognosis and seen in less than 10% of advanced ILD patients [10]. Development of pulmonary hypertension in ILD includes fibrosis in lung parenchyma causing pulmonary vascular bed destruction, leading to hypoxic pulmonary vasoconstriction and subsequent impairment of gas exchange, allowing for increase in pulmonary vascular resistance [11].

Pulmonary hypertension-related ILD is seen most commonly in IPF, chronic hypersensitivity pneumonitis, sarcoidosis, connective tissue disease-ILD, and pulmonary Langerhans cell histiocytosis [12]. Some ILD diagnoses have an overlap of both group 1 pulmonary arterial hypertension and group 3 PH, such as systemic sclerosis [13]. Furthermore, some conditions, such as sarcoidosis, pulmonary Langerhans cell histiocytosis and sickle cell disease, are classified as group 5 PH since causation is unclear or multifactorial [12].

## 3. Prognosis and Survival

The presence of pulmonary hypertension significantly worsens prognosis in patients with interstitial lung disease. ILD-PH is associated with a substantial increase in both morbidity and mortality, with affected individuals exhibiting accelerated functional decline, reduced exercise tolerance, and an elevated risk of right heart failure [14]. The mean survival for IPF patients with PH has been estimated to be around 2 years, significantly lower than the survival of IPF patients without PH, which ranges from 3 to 5 years (depending on disease stage), with right heart dysfunction being one of the main determinants of worse outcomes [15]. A more recent systematic review emphasized the significant association between PH and worse survival in ILD, with a pooled hazard ratio (HR) for mortality of approximately 2.5 [11]. Furthermore, IPF patients with PH have significantly worse survival compared with those with other types of ILD with PH [16] (see Figure 1). Overall, the presence of PH in ILD patients significantly worsens the prognosis, necessitating early screening, careful monitoring, and proactive management of both conditions in order to improve outcomes.

## 4. Clinical Presentation

Regardless of ILD classification, clinical presentation is similar, including insidious dyspnea on exertion, persistent dry cough, and/or fatigue [7]. As fibrosis progresses, dyspnea continuously and progressively occurs in addition to unintentional weight loss [2]. Development of pulmonary hypertension may be evident by complaints of lightheadedness and syncopal events with exertion, particularly as the right ventricle starts failing [5]. A physical exam reveals bibasilar dry inspiratory crackles and occasionally digital clubbing for pulmonary fibrosis and a parasternal heave, loud P2 heart sound, jugular venous distension, hepatomegaly, ascites, and peripheral edema in patients with severe pulmonary hypertension [5,6].

Practitioners should have a high index of suspicion for the association of pulmonary hypertension. Symptoms of shortness of breath due to underlying ILD can be confounded by other common comorbidities such as left heart disease, valvular heart disease, diastolic dysfunction, sleep apnea and associated alveolar hypoventilation, and chronic thromboembolic pulmonary hypertension.

## 5. Diagnostic Evaluation

Work-up requires a high-resolution CT scan (HRCT) as it shows visual evidence of lung inflammation and/or fibrosis, with characteristics including ground glass opacities, honeycombing, traction bronchiectasis, and/or subpleural reticulation. The pattern in which these findings are distributed may help dictate the underlying disease ideology. CT imaging of the chest revealing a right ventricle/left ventricle ratio greater than 1 or pulmonary artery/aortic ratio greater than 0.9 are suggestive of the presence of pulmonary hypertension. Patients should also have extensive laboratory testing to evaluate for any specific etiologies contributing to their ILD, including an autoimmune panel, hypersensitivity pneumonitis panel, and occasionally a genetic mutation panel if there is suspicion for familial pulmonary fibrosis. Multidisciplinary discussion with pulmonary medicine, radiology, pathology, and sometimes rheumatology is the gold standard approach for ILD diagnosis [9,17]. If there is still discrepancy regarding diagnosis after this approach, then a surgical lung biopsy may be considered at an expert center; however, TBLC has a 9% pneumothorax rate and 30% bleeding rate while surgical lung biopsy is invasive [9].

Determination of disease severity and ongoing disease monitoring includes the six-minute walk test distance and pulmonary function tests (PFTs), which may show a reduced total lung capacity, reduced forced vital capacity (FVC), and/or a reduced diffusing capacity for carbon monoxide (DLCO). However, in early disease, PFTs may be normal or only show mildly reduced DLCO. A reduction in DLCO (less than 30% predicted) out of proportion to parenchymal abnormalities may be suggestive of concomitant underlying pulmonary hypertension [18]. Frequent monitoring of PFTs at 3 to 6 month intervals should be pursued as even small declines in FVC of 5-10% at 6 months are associated with increased mortality, and greater FVC declines are associated with increased mortality [19,20].

Once the diagnosis of ILD is established, echocardiogram and plasma brain natriuretic peptide (BNP) or NT-proBNP should be pursued in all patients to screen for PH [6]. If there is suspicion for PH based on echocardiogram findings and/or elevated BNP or NT-proBNP, then right heart catheterization should be performed to confirm diagnosis and evaluate severity of pulmonary hypertension [6].

## 6. Therapeutic Options

For patients with inflammatory-predominant ILD, treatment depends on underlying disease etiology and typically includes immunosuppressant medications, such as glucocorticoids, mycophenolate mofetil, azathioprine, cyclophosphamide, anti-CD20 monoclonal antibody rituximab, and anti-IL-6 monoclonal antibody tocilizumab, among others. However, if fibrosis predominates, there is no role for anti-inflammatory therapy, and instead, antifibrotic therapy should be promptly initiated, including either nintedanib or pirfenidone for IPF, and nintedanib only for PPF.

The 2014 randomized, placebo-controlled, phase 3 ASCEND clinical trial compared thrice-daily oral medication pirfenidone, a transforming growth factor-ß inhibitor proposed to inhibit fibroblast proliferation and collagen synthesis, to placebo in IPF patients [21,22]. In ASCEND, the pirfenidone group had significantly slower disease progression, with reduction in FVC decline by 47.9% at one year, smaller decline in 6 min walk distances, and better progression-free survival, compared to the placebo group [22]. However, adverse events related to nausea and skin rash were more common in the pirfenidone group. Also published in 2014, INPULSIS-1 and INPULSIS-2 were two replicate phase 3 randomized, controlled clinical trials comparing twice-daily oral medication nintedanib, a tyrosine kinase inhibitor, to placebo in IPF patients [23]. Similar to ASCEND, the study found nintedanib had a similar slowing of FVC decline compared to placebo with notable adverse events of diarrhea [23]. The 2019 INBUILD randomized, placebo-controlled, phase 3 trial compared nintedanib to placebo in PPF patients, and also found a significant reduction in FVC decline in the nintedanib group, but with frequent diarrhea [24].

Since the approval of both nintedanib and pirefenidone by the Food and Drug Administration (FDA) in 2014, there have been no additional FDA approved medications for IPF or PPF. However, this is a potentially exciting time for treating a devasting disease as there is now greater understanding of the disease mechanisms. There is both epithelial and endothelial cellular injury within alveoli that lead to an exaggerated inflammatory response with sequelae of upregulation of myofibroblasts, allowing for excessive collagen deposition [7]. With improved knowledge of this fibrotic pathway, there are now several novel agents being actively studied in clinical trials to determine if various interruptions in this pathway will either slow down, halt, or reverse disease progression. Most promising is a phase III randomized, placebo-controlled clinical trial studying nerandimolast, a phosphodiesterase 4B inhibitor, (ClinicalTrials.govIdentifier: NCT05321082) which found that compared to placebo, this medication had a significant reduction in FVC decline at 52 weeks in IPF patients [25]. Currently this medication is awaiting FDA approval.

Inhaled treprostinil, a prostacyclin analogue that increases pulmonary vasodilation, is the only FDA-approved medication for group 3 PH-related ILD. The INCREASE trial, a randomized, double-blinded, placebo-controlled trial published in 2021, demonstrated an increased six-minute walk distance in those treated with inhaled treprostinil compared to placebo [26]. Although not statistically powered to do so, it also demonstrated an improvement in FVC in those treated with inhaled treprostinil. As a result, a 52-week randomized control trial called TETON is underway to determine the antifibrotic effects of treprostinil in patients with ILD.

There have been several negative clinical trials for various pulmonary vasodilator medications in PH-related ILD. Sildenafil has not shown any significant clinical benefit when used alone in patients with ILD. Sildenafil combined with nintedanib in the INSTAGE trial did not meet the primary endpoint of improved St George’s Respiratory Questionnaire with regard to health-related quality of life in patients with IPF [27]. Bosentan, a dual endothelin-1 receptor antagonist (ERA) that dilates the pulmonary vascular bed, did not improve pulmonary hemodynamics, six-minute walk distance, and symptoms, compared to placebo [28]. A randomized, double-blinded, placebo-controlled trial on ambrisentan, also an ERA, was terminated early as results suggested that it may increase risk for disease progression and respiratory hospitalization [29]. A randomized controlled trial comparing riociguat, a soluble guanylate cyclase stimulator, to placebo was also terminated early due to increased adverse events and mortality in the riociguat group [30].

In addition to medications, ILD management includes pulmonary rehabilitation, which is suggested to improve dyspnea, quality of life, and six-minute walk distance, and initiation of supplemental oxygen if indicated on testing, as this may improve breathlessness, quality of life, and mortality [31,32]. Attention to maintenance of weight is also key, as unintentional weight loss in ILD is associated with acute exacerbations and mortality [33,34].

However, despite these efforts, many patients with pulmonary fibrosis have ongoing disease progression, and acute exacerbations of pulmonary fibrosis are especially devastating with the 1-year incidence at 8–13% and a median survival of 2-4 months [35,36,37,38]. Therefore, pulmonary fibrosis is the most common indication for lung transplant in the United States, as it is the only curative option [39].

## 7. Gaps in Knowledge

There are still multiple gaps in our knowledge on how to best risk stratify and treat patients with PH-ILD. Despite multiple attempts in the literature, a validated risk assessment score for WHO group 3 PH-ILD is lacking. Like in WHO group 1 PH, a validated risk assessment score would replace the gestalt of the treating physicians with objective criteria when it comes to risk assessment and mortality. Secondly, there are no well-established protocols specific to the evaluation and treatment of PH-ILD. Much of this practice is borrowed from the WHO group I population and ESC/ERS guidelines, with the addition of clinical trials specific to ILD [10]. Thirdly, the clinical benefits regarding morbidity and mortality that result from disease-specific treatment of PH-ILD are not well described. Patients with fibrotic ILD collectively have an increased mortality. This is one of the most important reasons to identify and treat patients as early as possible, enroll patients in clinical trials, and refer to transplant centers.

## 8. Conclusions

ILD encompasses a wide variety of lung diseases leading to lung parenchymal inflammation and/or fibrosis. Although a rare disease, prevalence is rising with an aging population [1]. Development of fibrosis is a worrisome outcome with increased risk for respiratory failure and death [2]. The addition of pulmonary hypertension to pulmonary fibrosis further worsens outcomes, including increased risk of acute exacerbations, hospitalizations, and mortality [6]. Therefore, suspicion for ILD needs to be high in an aging patient with non-specific pulmonary symptoms, and once the diagnosis is established, screening for pulmonary hypertension must be pursued. Currently, treatment for both pulmonary fibrosis and group 3 PH-ILD is limited with only nintedanib or pirfenidone for the former, and inhaled Treprostinil for the latter. Therefore, early referral for lung transplantation is paramount until new effective agents are introduced.

## Figures and Tables

**Figure 1 biomedicines-13-00808-f001:**
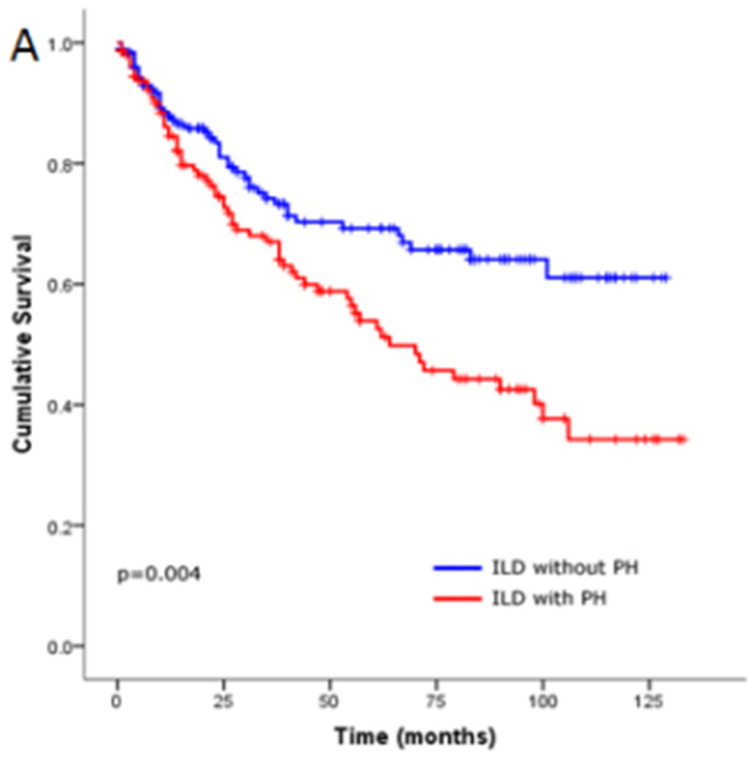
Kaplan–Meier survival estimates for the relationship with (A) interstitial lung disease (ILD) patients with and without pulmonary hypertension (PH).

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
