# Peer review of "Pulmonary Hypertension-Related Interstitial Lung Disease: An Expert Opinion with a Real-World Approach"

_biomedicines, 2025, doi:10.3390/biomedicines13040808_

Round 1

Reviewer 1 Report

Comments and Suggestions for Authors

This is a well written concise review on PH in ILD and provides readers with a good understanding of the topic. 

I have a few comments:

  1. Lines 55 - 59: The description of pathology/pathogenesis is a bit misquoted from the cited reference. The impairment of gas exchange (resulting from parenchymal destruction) leads to hypoxic vasoconstriction, which contributes to rise in PVR. The HPVC does not lead to subsequent impairment of gas exchange. This needs to be reworded. Also, I am not aware of any data suggesting endothelial cell proliferation. Mainly SMC and myofibroblasts. 
  2. Line 108: bronchoscopic lung biopsy would generally be contraindicated in PH. surgical would be high risk
  3. Line 170. there is increasing suggestion that sildenafil may be of value. See below refs: 
    1. PMID: 35452834
    2. PMID: 36172951

Author Response

  1. Lines 55 - 59: The description of pathology/pathogenesis is a bit misquoted from the cited reference. The impairment of gas exchange (resulting from parenchymal destruction) leads to hypoxic vasoconstriction, which contributes to rise in PVR. The HPVC does not lead to subsequent impairment of gas exchange. This needs to be reworded. Also, I am not aware of any data suggesting endothelial cell proliferation. Mainly SMC and myofibroblasts.  Response: I agree. I deleted the sentence. 
  2. Line 108: bronchoscopic lung biopsy would generally be contraindicated in PH. surgical would be high risk. Response. I agree. I deleted the transbronchial biopsy and left a surgical biopsy.
  3. Line 170. there is increasing suggestion that sildenafil may be of value. See below refs: 
    1. PMID: 35452834
    2. PMID: 3617295 Response. I agree. Point well taken. These were not mentioned in an attempt to not be misleading  as sildenafil is not FDA approved for WHO group 3 PH. 

Reviewer 2 Report

Comments and Suggestions for Authors

Rachel N. Criner et al. submitted an interesting paper about the interplay between LH and ILD. The topic was less frequently reported yet of a certain significance. It was believed that the submission fell within the scope of Biomedicines. The reviewer suggested a Minor Revision for this paper. Detailed comments:

  1. The Abstract contained about ~120 words, which was a bit short. Please consider to expand it to showcase more valuable information.
  2. As the definition and statistics of ILD were provided in the Introduction, it was also recommended to add those of PH.
  3. A separate paragraph to describe the framework of this paper should be supplemented at the end of the Introduction.
  4. In the Section of Prognosis and Survival, some relevant data might be shown, especially some figures or tables.
  5. NCT number of the clinical trials in Section 6 should be provided.

Besides, in the SuSy system, the article type of the submission was Opinion, while in the PDF file it was Review. Please double-check this issue.

Author Response

  1. The Abstract contained about ~120 words, which was a bit short. Please consider to expand it to showcase more valuable information.
  2. As the definition and statistics of ILD were provided in the Introduction, it was also recommended to add those of PH.

Response: Combined recommendations for comments 1 and 2 :

There has been great progress in the treatment of pulmonary arterial hypertension (WHO group 1 PAH) over the past two decades which has significantly improved the morbidity and mortality in this patient population.   Likewise, the more recent availability of antifibrotic medications for interstitial lung disease (ILD) have also been effective in slowing down the progression of disease. There is no known cure for either of these disease states.  When this combination coexists, treatment can be challenging. Interstitial lung disease is a heterogenous group of chronic inflammatory and/or fibrotic parenchymal lung disorders.  A subset of patients with ILD, not related to connective tissue disease, can initially present with inflammatory-predominant disease which progresses to irreversible fibrosis. This population of patients is also at risk for developing pulmonary hypertension (PH) or World Health Organization (WHO) group 3 PH. This coexistence of ILD and PH is associated with early morbidity and mortality.  The early identification, diagnosis and treatment of this combination of ILD and PH is vital.  Medications available for both ILD and PH require an individualized approach with the intention of slowing down disease progression. Referral to expert centers for clinical trials and transplant evaluation is recommended.

3.  A separate paragraph to describe the framework of this paper should be supplemented at the end of the Introduction.

Response: 

The combination of PH-ILD can be challenging to diagnose and treat effectively. Patients require a thorough clinical evaluation to enable the most accurate diagnosis. A vital part of that evaluation is the early recognition of PH.  Medications can help improve disease progression along with clinical trials that will further improve our gaps in knowledge.  

4. In the Section of Prognosis and Survival, some relevant data might be shown, especially some figures or tables.

 Response: e0141911. doi:10.1371/journal. 

Isert the graph from page 7 of this article that is sited in the paragraph.

5.  NCT number of the clinical trials in Section 6 should be provided

Response:  ClinicalTrials.govIdentifier: NCT05321082